# Monoclonal Antibodies as SARS-CoV-2 Serology Standards: Experimental Validation and Broader Implications for Correlates of Protection

**DOI:** 10.3390/ijms242115705

**Published:** 2023-10-28

**Authors:** Lili Wang, Paul N. Patrone, Anthony J. Kearsley, Jerilyn R. Izac, Adolfas K. Gaigalas, John C. Prostko, Hyung Joon Kwon, Weichun Tang, Martina Kosikova, Hang Xie, Linhua Tian, Elzafir B. Elsheikh, Edward J. Kwee, Troy Kemp, Simon Jochum, Natalie Thornburg, L. Clifford McDonald, Adi V. Gundlapalli, Sheng Lin-Gibson

**Affiliations:** 1Biosystems and Biomaterials Division, National Institute of Standards and Technology (NIST), Gaithersburg, MD 20899, USA; jerilyn.izac@nist.gov (J.R.I.); adolfas.gaigalas@nist.gov (A.K.G.); linhua.tian@nist.gov (L.T.); elzafir.elsheikh@nist.gov (E.B.E.); edward.kwee@nist.gov (E.J.K.); 2Applied and Computational Mathematics Division, National Institute of Standards and Technology (NIST), Gaithersburg, MD 20899, USA; paul.patrone@nist.gov (P.N.P.); anthony.kearsley@nist.gov (A.J.K.); 3Abbott Laboratories, Abbott Park, IL 60064, USA; john.prostko@abbott.com; 4Laboratory of Pediatric and Respiratory Viral Diseases, Office of Vaccines Research and Review, Center for Biologics Evaluation, Food and Drug Administration (FDA), Silver Spring, MD 20993, USA; hyungjoon.kwon@fda.hhs.gov (H.J.K.); weichun.tang@fda.hhs.gov (W.T.); martina.kosikova@fda.hhs.gov (M.K.); hang.xie@fda.hhs.gov (H.X.); 5Vaccine, Immunity and Cancer Directorate, Frederick National Laboratory for Cancer Research (FNLCR), Frederick, MD 21702, USA; kemptj@mail.nih.gov; 6Roche Diagnostics GmbH, 82377 Penzberg, Germany; simon.jochum@roche.com; 7Centers for Disease Control and Prevention (CDC), Atlanta, GA 30329, USA; nax3@cdc.gov (N.T.); ljm3@cdc.gov (L.C.M.); ibk8@cdc.gov (A.V.G.)

**Keywords:** SARS-CoV-2, spike protein, serological binding assay, neutralization assay, WHO international standard (WHO IS), monoclonal antibody, normalization, harmonization, uncertainty quantification, result comparability and traceability

## Abstract

COVID-19 has highlighted challenges in the measurement quality and comparability of serological binding and neutralization assays. Due to many different assay formats and reagents, these measurements are known to be highly variable with large uncertainties. The development of the WHO international standard (WHO IS) and other pool standards have facilitated assay comparability through normalization to a common material but does not provide assay harmonization nor uncertainty quantification. In this paper, we present the results from an interlaboratory study that led to the development of (1) a novel hierarchy of data analyses based on the thermodynamics of antibody binding and (2) a modeling framework that quantifies the probability of neutralization potential for a given binding measurement. Importantly, we introduced a precise, mathematical definition of harmonization that separates the sources of quantitative uncertainties, some of which can be corrected to enable, for the first time, assay comparability. Both the theory and experimental data confirmed that mAbs and WHO IS performed identically as a primary standard for establishing traceability and bridging across different assay platforms. The metrological anchoring of complex serological binding and neuralization assays and fast turn-around production of an mAb reference control can enable the unprecedented comparability and traceability of serological binding assay results for new variants of SARS-CoV-2 and immune responses to other viruses.

## 1. Introduction

Coronavirus disease 2019 (COVID-19) caused by severe acute respiratory syndrome coronavirus 2 (SARS-CoV-2) has resulted in unprecedented disruptions to society, but also led to impressive innovations in the fields of diagnostics, vaccines, and therapeutics. Serology assays have been and continue to be vital for managing COVID-19 [1,2]. Neutralizing antibody titers have been widely used to assess the vaccine efficacy for immunological correlates of protection (CoPs) for investigational and licensed vaccines, including the most recent bivalent boosters [3,4]. In this context, a CoP is defined as an immune marker that can be used to reliably predict a vaccine’s efficacy in preventing a clinically significant outcome [5,6,7]. Anti-SARS-CoV-2 spike antibody levels can also be used for establishing CoPs, as supported by animal studies [8,9], natural infection cohorts [10], and vaccine trial studies [11]. The US Food and Drug Administration (FDA) and European Medicines Agency (EMA) have since accepted these two CoPs, anti-spike-binding antibody titer and neutralization antibody titer for vaccine assessment and approval [7].

Serological binding assays are known to have large variations across different laboratories due to the multi-component nature of the assays [2,12]. Among the SARS-CoV-2 serological binding assays, immobilized recombinant antigens, either the full spike protein, receptor-binding domain (RBD), subunit 1 (S1) of the spike protein, or nucleocapsid protein (N), have been utilized for detecting different isotypes of antiviral antibodies with different detection modalities [13]. A binding assay can be designed using different formats, such as bead-based and plate-based, further introducing measurement variabilities [14]. These complexities have made it difficult to evaluate assay accuracy, precision, robustness and compare results obtained from different assays. The FDA removed 27 serology tests from its Emergency Use Authorization (EUA) in May 2020 due to the lack of assay validation data and potential risks to public health. In response, the World Health Organization (WHO), in collaboration with the National Institute for Biological Standards and Control (NIBSC) initiated an effort to develop the first WHO international standard (WHO IS) and reference panel for anti-SARS-CoV-2 antibody for normalizing serological assays using a pool of plasma from 11 SARS-CoV-2 convalescent patients. Since the establishment of the WHO IS in November 2020 [15], several studies were conducted for evaluating its suitability for COVID-19 serology assay [16,17,18]. The conclusion of these investigations was that normalization using the WHO IS improved analytical and diagnostic comparability by placing results on a similar scale, but did not remove any sources of variability, including those associated with different assay reagents and platforms/formats. Importantly, the lack of uncertainty quantification hindered assay harmonization and the quantitative assessment of assay comparability. In addition, the rapid uptake of this standard led to a depletion of stock by August 2021. The establishment of the second WHO IS for anti-SARS-CoV-2 immunoglobulin and a reference panel for antibodies to SARS-CoV-2 variants of concern [19] that occurred in July 2022 and the subsequent expansion of the WHO reference panel for antibodies to SARS-CoV-2 variants of concern [20] that took place in March 2023 more directly addressed the reference standard needs for the recently emerged and widely circulated Omicron variants.

Due to biological complexities, the standardization of serology assays has been extremely challenging [21]. It is generally accepted that only matrix-matched and pooled convalescent reference standards have the breadth of epitopes necessary for standardizing various serology assays. However, to the best of our knowledge, there are no definitive data supporting this claim for SARS-CoV-2. Moreover, the curation, pooling, preparation, and testing of large volumes of convalescent patient samples is a time-consuming process, as shown by the WHO standardization effort. As SARS-CoV-2 continues to rapidly evolve [22], it becomes even more challenging to keep pace with the most current variants to obtain a pure convalescent sample from a specific viral variant serving as an assay reference standard for the variant-specific assay. In addition, pooled samples are exceedingly complex; so, bridging and/or traceability concepts are nearly impossible to implement.

Monoclonal antibodies (mAb), including anti-SARS-CoV-2 spike-neutralizing antibodies (nAb), can be generated rapidly with the current technologies [3,23]. mAbs have been used extensively as in-house reference materials for serology but have not been considered for broad use. In this paper, we conduct a small-scale interlaboratory serology study consisting of WHO IS, three anti-SARS-CoV-2 spike mAbs, and 62 human serum/plasma samples to (1) evaluate the suitability of an mAb panel to enable comparability and standardized results of serological binding and neutralization assays, (2) understand the strengths and weaknesses of different neutralization assay formats, and (3) establish more predictive methods for correlated protection for SARS-CoV-2.

## 2. Results

### 2.1. Serological Binding Assays

Figure 1a shows the ‘Day 1’ measured dilution curves for all 62 samples along with the WHO IS obtained by one lab. Figure 1b illustrates the corresponding data collapse onto the dilution curve of the WHO IS. Note that we rotated the mean fluorescence intensity (MFI)readout in Figure 1a to the horizontal axis of Figure 1b to streamline the normalization algorithm [24]. In Figure 1a, and as expressed by Equation (4), the magnitude of the translation factor on the log scale for each curve relative to that of the standard ${\ln(\alpha }_{s,n,r})$ is equal in magnitude to the scaled antibody concentration $\gamma_{s,n,r}.$ This new analysis method inherently reduces many sources of variability, and surprisingly, we were able to determine $\gamma_{s,n,r}$ values with precise uncertainty quantifications for what is generally thought to be semi-quantitative assays. We repeated this process for all combinations of standards and assays. The average log antibody concentration, ${\bar{\gamma }}_{s,n,r},$ determined from $\gamma_{s,n,r}$ measured on separate days, was used in all subsequent analyses.

Figure 2 shows the results from all participants normalized and harmonized to the WHO IS using the new analysis methods described above (Equations (4)–(6)), where the samples are ordered from the lowest to largest value of log antibody consensus concentration. The average of each log concentration $\ln({\hat{c}}_{s,n,r})$ normalized to the WHO IS clearly shows a systematic bias for each assay relative to the others that corresponds to the systemic bias $\Delta g_{r,n}$ (Figure 2a). The consensus antibody concentrations c_s,r_ determined using Equation (5) are shown as red squares along with harmonized antibody concentrations $H_{s,r}$, as shown in Figure 2b. The removal of contributions from $\Delta g_{r,n}$ clearly improved agreements among all laboratories, with the results better coalescing around c_s,r_. The randomness associated with assay- and sample-dependent uncertainties $\Delta g_{s,n}$ and $\delta_{s,n}$, respectively, were quantified, where differences from different laboratories/methods are clear (Figure 2c).

When the same analysis procedure was implemented using each of the three mAb as the standard, we observed the following: identical trends for ${\ln(\hat{c}}_{s,n,r})$, a slightly different value of c_s,r_ due to the reference-dependent bias $\Delta g_{r,n}$, and identical values for random assay- and sample-dependent uncertainties (Appendix C, Figure A1), all consistent with the predictions from the above equation and our theory [24]. Figure 3 summarizes the log consensus values for each sample using the different standards. Note that, while the consensus values for a fixed sample differ according to standard, the difference is sample-independent. In other words, the consensus values for all standards are interchangeable up to a constant, which is straightforward to determine using any sample. This ambiguity in the definition of the consensus value arises from the fact that only differences between Gibbs free energies are meaningful, not the free energies per se.

### 2.2. Neutralization Assays

The NT50 values were determined using the Hill equation described in the ‘Centralized Data Analysis’. Figure 4 shows the results of the Wilcoxon matched-pairs signed-rank tests [25] performed on the bead-based surrogate, pseudovirus-based neutralization assay (pvNT), and the live-virus microneutralization assay (MN). Each assay pair was significantly different. Based on a limit of detection (LOD) of 8.94 for the surrogate assay determined using the 24 negative samples, the surrogate assay identified 28 of 39 positive serum samples, including the WHO IS (Figure 4a,c). The Spearman correlation was 0.928, *p* < 0.000, between pvNT and the surrogate assay (Figure 4a); 0.866, *p* < 0.0001, between pvNT and the MN assay (Figure 4b); and 0.738, *p* < 0.0001, between MN and the surrogate assay.

We further compared the three different neutralization assays using receiver operating characteristic (ROC) curves to evaluate the performance of the assays (Figure 4d) [26]. The optimal threshold for pvNT was an NT50 of 2.00, yielding a sensitivity of 87.2% and a specificity of 100%. There were no false positives reported, but some false negatives. For the surrogate assay, the optimal threshold based on the ROC curve was an NT50 of 8.87, with a sensitivity of 70.3% and a specificity of 92.3%. This optimal threshold is in good agreement with the LOD of 8.94 noted above. Lastly, the ROC analysis led to a determination of an optimal NT50 threshold of 47.50, with a sensitivity of 86.6% and a specificity of 79.2%, for MN. The values of the area under curve (AUC) were 0.931 for pvNT, 0.888 for the surrogate assay, and 0.885 for MN. A higher AUC indicates that an assay is better able to identify neutralization.

### 2.3. Probability for CoPs

Considering pvNT as an optimal predictor of the presence of the neutralizing antibodies in this study, we constructed the probability model connecting consensus-binding concentrations to their neutralization counterparts. The left plot of Figure 5 shows the relationship expressed by Equation (8) for mAb 1 vs. the NT50 values from pvNT. It is notable that the mean relationship is not linear. The physical origins of this discrepancy are likely associated with the complexity of biological responses and point to the differing information content provided by the binding and neutralizing assays.

From a practical standpoint, however, the relationship expressed by Equation (8) and illustrated in Figure 5 (left) still provides useful information. For example, given a desired NT50 level $\nu_{min}$, we can determine the minimum consensus-binding concentration that would yield a neutralization measurement $\nu \geq \nu_{min}$ with a probability of 95%. Figure 5 (right) provides a probability function where the vertical line indicates that a log consensus-binding level of ~8.1 yields a >95% chance that the neutralizing titer is greater than 40 (exp(3.68)). While not pursued in this work, it is possible to incorporate the residual uncertainty associated with the variance of $\Delta g_{s,n}$ to estimate the harmonized binding level from a given assay for which that corresponds to $\nu \geq \nu_{min}$ with a desired probability. Such tasks are assay-specific and will be addressed in future work. In either case, however, it is straightforward to show that changing the reference material only rescales the coefficients associated with the deterministic part of Equation (8) but does not otherwise modify the probability model associated with the plot shown in Figure 5 (left). Thus, a decision based on the correlates of protection, as we defined, is also invariant to the reference material used.

## 3. Discussion

### 3.1. Serological Binding Assays

The results shown in Figure 2 and Figure A1, together with the corresponding theories [24], clearly demonstrate that the choice of reference does not affect the harmonized antibody concentration $H_{s,r}$. In other words, all binding references performed equivalently for the purposes of harmonization when using our thermodynamically derived analysis, as shown in Figure 3, successfully meeting the first objective of the interlaboratory study. This critical finding will allow a more rapid development of an mAb-based reference standard to support assay comparability, particularly during the early onset of an outbreak. As we witnessed for COVID-19, the development of pooled human serum reference materials was a lengthy effort that cannot keep pace with the rapidly evolving virus. Moreover, our analyses enable the quantitatively bridging of different reference standards, since their respective, harmonized concentrations only differ by constant biases associated with $\Delta g_{r,n}$. This aspect of our approach is extremely useful if multiple standards are developed and deployed by the community, as in the case of COVID-19.

Our analysis examines the choice of reference through the lens of uncertainty quantification (UQ). Specifically, our new analysis method (Equation (5)) shows that the average log antibody concentration ${\bar{\gamma }}_{s,n,r},$ an experimentally determined dimensionless value, is composed of four components: consensus antibody concentration c_s,r_, assay- and reference-dependent (systematic) bias $\Delta g_{r,n}$, and assay- and sample-dependent uncertainties $\Delta g_{s,n}$ and $\delta_{s,n}$, respectively. The contributions od $\Delta g_{r,n}$ and $\Delta g_{s,n}$ are important because c_s,r_ cannot be determined in a diagnostic setting where only one assay is used. Thus, the harmonized measurement $\gamma_{s,n,r}-\Delta g_{r,n}$ is our best estimate of c_s,r_, and the variance $\varsigma_{n}^{2}$ of $\Delta g_{s,n}$ quantifies the extent to which sample–assay effects render the consensus unknown.

Figure 6 (left) examines the combined contribution from $\Delta g_{r,n}$ and $\Delta g_{s,n}$ in absolute values. Note that thermodynamics underpinning this led to the opposite sign of $\Delta g_{r,n}$ and $\Delta g_{s,n}$ in Equation (5); this means that their contributions can be canceled out if an improper experimental design is conducted when developing a reference. A comparison of the $\Delta g_{r,n}$ value determined when using the WHO IS versus an mAb provides a quantitative measure of the additional variability introduced when an mAb is used as the normalization standard (not the harmonization standard). In essence, mAbs can be used as a normalization standard, albeit with increased (~8x) variabilities as compared to WHO IS, consistent with the notion that a pooled sample serves as a better standard for normalization. However, this variation $\Delta g_{r,n}$ is all due to bias, which can be quantitatively removed, leaving $\Delta g_{s,n}$ as the only thermodynamic source of variability and therefore the identical performance of mAb and pooled convalescence standard via harmonization. Figure 6 (right) shows the per-lab estimate of the standard deviation in $\Delta g_{s,n}$ computed using each of the reference materials, an expansion of the orange portion of the Figure 6a that can be interpreted as the uncertainty in harmonized (i.e., bias-corrected) concentrations across the samples. Importantly, $\varsigma_{n}^{2}$ does not change (to within statistical uncertainty) according to the reference material, further supporting the notion that random bias arises from the samples and not from the reference interaction with the assay. By comparison, Labs 2, 3, and 4 demonstrated exceptional robustness/precision.

One unexpected finding is that, through the new analysis methods, we were able to generate quantitative metrics, namely, c_s,r,_ $\Delta g_{r,n}$, and the variance $\varsigma_{n}^{2}$ associated with $\Delta g_{s,n}$ for all assays, including those typically considered qualitative. This reflects the fact that, through the collection of a larger dataset to generate a master curve from sample dilution curves (Figure 1), not just a small linear range of sample dilution curves, a common practice in serology, we can reduce and/or remove the large bias arising from any given data point. This further supports the strengths of our method in normalizing and harmonizing a more diverse set of assay formats and predicting robust assays through uncertainty quantification.

### 3.2. Neutralization Assays

Our interlaboratory study includes assays spanning live-virus MN assay to surrogate the neutralization assay, enabling the assessment of the strengths and weakness of different neutralization assay formats. Live-virus MN assays are thought to provide the most biologically relevant data but must be conducted in a biosafety level 3 (BSL3) containment facility. They are also time-consuming, labor-intensive, and known to have relatively large uncertainties [27]. Hence, pvNT and non-cell based surrogate assays that can be performed in a BSL2 facility have gained enormous attraction. pvNT arguably mimics the live-virus MN assay better, whereas non-cell-based surrogate assays are inherently better controlled with an expected higher reproducibility. However, there are limited data that directly compares the performance of different neutralization assays formats [28]. Since we have results from each assay category, we can direct compare these. Note that, since neutralization studies were not performed in the same manner as serological anti-spike assays, we could not use the thermodynamic-based harmonization analysis method for centralized analysis.

The surrogate assay clearly showed the highest precision (Figure 4). Consistent with previous reports, the MN assay showed large uncertainties. By the ROC analysis, the pvNT and MN assays show similar sensitivity, 87.2% and 86.6%, respectively, indicating pvNT is a good alternative to the more complex MN assay. The faster surrogate assay showed the lowest sensitivity (70.3%). Interestingly, the specificity of pvNT was 100%, followed by the surrogate assay at 92.3%. MN showed the lowest specificity at 79.2%, indicative of the significant number of false positives detected, perhaps due to cross-reactivity. Although pvNT has the lowest detection threshold based on the ROC curve (label not visible), it is accepted that pvNT will unlikely reach 100% sensitivity because the selected COVID-19-positive (determined by PCR analysis) samples may, in fact, contain a low IgG titer. The lower sensitivity of the RBD-based surrogate assay may also be due to contribution from the N-terminal of the spike protein other than the RBD to prevent the virus’s entry into the target cells [29]. From the ROC curve analyses and AUC values, pvNT emerges as the best-performing neutralization assay for this study, although we note the choice of assay should ultimately be selected based on fit for purpose.

### 3.3. Correlates of Protection (CoPs)

A fundamental goal of serology testing is to understand the extent to which an individual is protected from further infection. Because neutralization assays are essential for determining the infection rate, predicted humoral protection, as well as vaccine efficacy during clinical trials and after large-scale vaccination, the different platforms of neutralization assays, including MN, pvNT, and surrogate assay, have been developed [30]. An increasing body of literature supports the neutralization assay and the clinical outcome. In this section, we make several observations and conclusions regarding the use of serological assays as CoPs on the basis of our work.

Chief among these is the recognition that assessing immunity, whether expressed in NT50 levels or otherwise, amounts to decision making under uncertainty. This is most clearly expressed by Figure 5, which shows that consensus-binding measurements are an imperfect proxy for neutralization levels. If we wish to deduce the latter from the former, we incur a penalty in the form of increased uncertainty in the NT50 value. This observation holds more generally when we replace one measurement with another, as is often the case when a gold standard is expensive to deploy but a surrogate is not. A key goal of our analysis has therefore been to estimate this uncertainty “penalty”. This can be used to make quantitative statements to the effect that a certain binding level ensures a minimum needed neutralization level with sufficient probability (e.g., 95%). It is likely that such probabilistic statements are the most practical and accessible routes to supporting decision-making processes in diagnostics.

This approach to UQ-based decision making is also well adapted to incorporating information from our harmonization analysis. As discussed in previous sections, the uncertainty $\Delta g_{s,n}$ quantifies the extent to which the consensus value is unknown, given the harmonized (i.e., bias-corrected) measurement associated with a specific assay. Through well-known uncertainty propagation techniques, the variance $\varsigma_{n}^{2}$ can be incorporated into the uncertainty model associated with Equation (8) and the binding–neutralization relationship. This allows for a decision to be made directly on the basis of a specific serology measurement, not just a consensus concentration, which is in general unknown; see also ref. [24] for more details. In this way, the UQ also provides a route for determining which binding assay is a best proxy for neutralization measurements. Moreover, it facilitates downstream analyses that can connect uncertainty associated with neutralization measurements to more real-world measures of protection, e.g., associated with clinical outcomes and individual risk levels.

## 4. Materials and Methods

### 4.1. Interlaboratory Study Design

The study materials consisted of WHO IS, 3 anti-SARS-CoV-2 spike mAbs, and 62 human serum/plasma samples. Two of the three mAbs were provided by Regeneron via a material transfer agreement. Another mAb and 47 convalescent serum samples consisting of 24 negative samples (collected pre-pandemic) and 23 positive samples (natural infection verified by PCR tests) were donated by Abbott Laboratories. Fifteen convalescent plasma/serum samples provided by CDC were from vaccinated individuals, and five samples from each vaccine modality: Johnson and Johnson, Moderna, and Pfizer. All convalescent sample curations were IRB-approved by the respective organizations. The study was also approved by the IRB office of the National Institute of Standards and Technology (NIST) through an individual material transfer agreement with each participating organization.

A detailed study design is provided in the Appendix A. A study protocol was provided to participants to specify the requirements for preparing and handling samples and controls. A reporting template was provided to harmonize the reporting of assay information and results.

Six serological binding assays and three neutralization assays were incorporated in the centralized data analysis of this interlaboratory study (see Table 1 and Appendix B for more information).

### 4.2. Centralized Serology Data Analysis

Data analysis proceeds via a hierarchy of methods motivated by concepts from thermodynamics and conditional probability. A core idea of this analysis is that normalization and harmonization are distinct tasks. A complete justification is provided in our companion paper [24]. A brief explanation is provided below.

Serology assays always involve a reversible binding interaction of the form,
(1)Y+B↔YB=C
where *Y* is an antibody, *B* is a substrate, and C is an antibody–substrate complex. When used as a measurement tool, such assays quantify not only the properties of the antibodies but also the substrate, e.g., binding for a given epitope. Stated differently, serology measures the properties of a molecular interaction, in this case, the equilibrium binding between antibody in blood serum and the antigen-bound substrate. As such, it is impossible to fully remove effects of the substrate from the process of quantifying antibody concentration. This means the measurement does not determine the reactants or antibody concentrations, even though the serology results are generally reported in terms of antibody titer.

This observation motivated us to define normalization as the task of estimating the antibody quantity relative to a chosen reference for a fixed assay. The normalized antibody concentration ${\hat{c}}_{s,n,r}$ is
(2)$${\hat{c}}_{s,n,r}={\hat{y}}_{s,n,r}B$$
where ${\hat{y}}_{s,n,r}$ is the dimensionless, scaled antibody concentration of sample *s* relative to reference *r* using assay *n* (Equation (3)) and *B* is an arbitrarily assigned binding antibody unit (BAU) concentration, e.g., associated with the WHO IS or another standard. Here and throughout the paper, the subscripts *s*, *n*, and *r* are reserved for a sample, assay, and reference, respectively. ${\hat{y}}_{s,n,r}$ is defined as
(3)$${\hat{y}}_{s,n,r}=\frac{c_{s,n}}{c_{r,n}}\;»\;\frac{{MFI}_{s,n}}{{MFI}_{r,n}}\;\mathrm{or}\;\frac{{OD}_{s,n}}{{OD}_{r,n}},$$
where $c_{s,n}$ and $c_{r,n}$ are the absolute concentrations of bound antibodies associated with the sample and reference, respectively. Note that the reference is generally a calibrator for the specific assay. In practice, ${\hat{y}}_{s,n,r}$ is generally calculated via the ratio of mean fluorescence intensity or optical density (OD) in the linear range of the log–log curve of serology assays.

To better utilize the entire set of data from the dilution measurements that reduces random variations, we developed a novel analysis method that generalizes the slope-correction method [31] and directly estimates ${\hat{y}}_{s,n,r}$
(4)$${\hat{y}}_{s,n,r}={\hat{\alpha }}_{s,n,r}^{-1}$$
where $\alpha_{s}$ is the translation factor for each sample, such that the fluorescence values $f(\alpha_{s}d)$ as a function of dilution d all collapse onto the master curve (see Appendix A). In other words, the scaling factor $\alpha_{s,n,r}$ laterally translates the dilution curve when considering fluorescence as a function of $\gamma_{s,n,r}=\ln({\hat{y}}_{s,n,r})$, where $\gamma_{s,n,r}$ is the log-scaled antibody concentration.

In contrast, harmonization is intuitively defined as the process of altering the numerical values of normalized concentrations so that the (normalized) measurements of each assay become interchangeable for a fixed sample. This leads to the concept of a consensus antibody concentration $c_{s,r}$ for each sample, to which all harmonized measurements are sufficiently close, i.e., within a quantifiable uncertainty. More specifically, thermodynamic arguments imply that the experimentally determined, average (e.g., as measured over multiple days), log-scaled antibody concentration ${\bar{\gamma }}_{s,n,r}$ is comprised of four components according to
(5)$${\bar{\gamma }}_{s,n,r}{=\ln(c}_{s,r})+\Delta g_{r,n}-\Delta g_{s,n}+\delta_{s,n}$$
where $\Delta g_{r,n}$ and $\Delta g_{s,n}$ are Gibbs free energies associated with the reference–assay pair and sample–assay pair, respectively, and $\delta_{s,n}$ is a sample-dependent day-to-day variation associated with instrument noise, operator effects, etc. [32]. Importantly, $\Delta g_{r,n}$ introduces a systematic bias that depends on the choice of the reference, whereas $\Delta g_{s,n}$ introduces random bias, expressed as variance $\varsigma_{n}^{2}$. Note that, although the values of $c_{s,r}$ depend on the reference used, they are readily interchangeable among different references.

Equation (5) provides the bases for determining the harmonized antibody concentration, $H_{s,r}$, via
(6)$$\ln(H_{s,r})={\bar{\gamma }}_{s,n,r}-\Delta g_{r,n}=\ln(c_{s,r})-\Delta g_{s,n}+\delta_{s,n}$$

Effectively, $H_{s,r}$ is the reference–assay dependent, systematic-bias-corrected, averaged log antibody concentration or the consensus antibody concentration together with the random assay–sample-dependent uncertainties. Equation (6) is key to understanding the uncertainties from a given serology assay.

### 4.3. Neutralization Assays

For serology measurements, diluting samples containing antiviral antibodies changes the number of antibodies bound to the assay surface, as given by the Hill equation, shown in Equation (7) [33].
(7)$$N_{Ab}=N_{max}+\frac{N_{min}-N_{max}}{\left(1+{\left(\frac{{NT}_{50}}{dil}\right)}^{n}\right)}$$
where *N_Ab_* is the total number of labeled Ab immobilized on surface and *N_max_* and *N_min_* are the maximum and minimum numbers of the labeled *N_Ab_*, respectively. *NT*_50_ equals to the dilution at which half of the immobilized antigen molecules are filled and *n* is the Hill coefficient. Equation (7) is an excellent representation of Gibbs free energy function characterizing the thermodynamic interaction system between immobilized antigen molecules and antiviral antibodies in the sample. A detailed procedure for applying Equation (7) for the three neutralization assays platformed is provided in the Appendix A.

### 4.4. Probability Models for CoPs

To construct the probability models that connect binding and neutralizing assays, we used the consensus values $\chi_{s,r}$ for a fixed reference and the harmonic mean (across days) of NT50 values $\nu_{s}$ associated with a single neutralizing assay. It stands to reason that, on average, increasing consensus values should correspond to increasing NT50 estimates. However, given the uncertain biochemical connection between binding measurements and their neutralization counterparts, we anticipated that the relationship between them is (i) not necessarily linear and (ii) most likely partly random.

This motivated us to postulate a minimal model in which $\ln(\nu_{s})$ is a low-order polynomial of $\chi_{s,r}$ with a constant-variance Gaussian noise. More precisely, we posited that
(8)$$\ln\left(\nu_{s}\left(\chi_{s,r}\right)\right)=a_{0}+a_{1}\chi_{s,r}+a_{2}\chi_{s,r}^{2}+\epsilon_{neut}$$
where the $a_{0},\;a_{1},$ and $a_{2}$ are unknown parameters and $\epsilon_{neut}$ is a normal random variable with an unspecified mean and variance. We determined these unknown parameters using the maximum likelihood analysis.

### 4.5. Disclaimer 

To specify an experimental procedure as completely as possible, certain commercial materials, instruments, and equipment were identified in this manuscript. In no case does the identification of the manufacturer of particular equipment or materials imply a recommendation or endorsement by the National Institute of Standards and Technology or the Centers for Disease Control and Prevention, nor does it imply that the materials, instruments, and equipment identified are necessarily the best available for the purpose. The findings and conclusions in this article are those of the authors and do not necessarily represent the views of the Centers for Disease Control and Prevention or the Agency for Toxic Substances and Disease Registry.

## 5. Conclusions

The results from this interlaboratory study and accompanying new analysis methods provide a means to achieve unprecedented serological-binding assay harmonization. Importantly, our analysis methods afford a more rigorous quantitation of antibody concentrations as well as sources of uncertainties, thus allowing a better understanding of assay performance. Generally, harmonization is sought through experimental design that requires all laboratories use the same reagents to perform the identical assay. Because of logistics and supply chain issues associated with this method of assay harmonization and the existence of diverse serological assay platforms using different assay reagents during the COVID-19 pandemic, the harmonization of serological-binding assays has not been accomplished at present. In this paper, we conclusively demonstrated that a single anti-spike mAb can be used alone for assay harmonization when utilizing the new analysis method. Considering the presence of different viral variants respective to the original Wuhan-Hu-1 strain investigated in this study, a cocktail of mAbs with different binding specificities to different viral strains would be better to harmonize serological-binding assays across different viral strains. This means that mAb-based reference materials can be rapidly developed to support emerging infections. Together, and for the first time, we can establish methods to enable traceability and comparability across different assay platforms and reference materials.

This interlaboratory study also directly compared surrogate, pvNT, and MN neutralization assays and showed that the pvNT assay had the highest sensitivity and specificity compared to the other methods. Using NT50 values from the pvNT assay, we showed good correlations with binding assays used in the study, consistent with literature reports of the two markers, anti-spike antibody titer and neutralizing antibody titer, serving as CoPs for the efficacy evaluation of vaccines and COVID-19 disease management.

This approach was developed and demonstrated for COVID-19, although we expect the thermodynamics underpinning of binding assays to be generally applicable to broader serological assays, including efforts to develop new vaccines for a variety of diseases, such as RSV and pan-influenza, for seroprevalence monitoring and in the assessment of pre-existing immunity prior to the administration of therapeutics, such as emerging gene therapies.

## Figures and Tables

**Figure 1 ijms-24-15705-f001:**
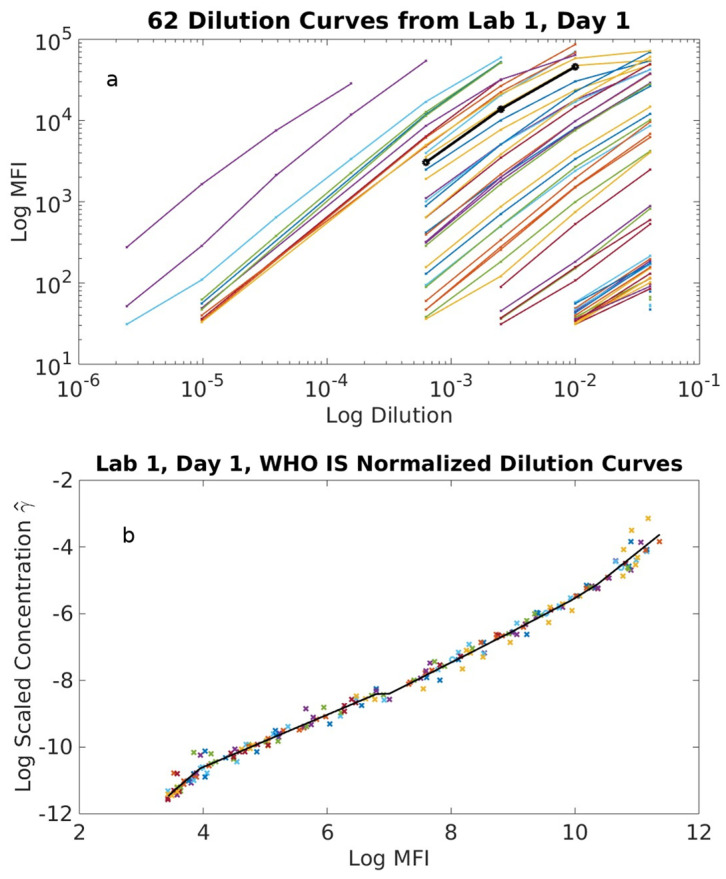
(**a**) Sixty-two sample dilution curves from experiment Day 1 obtained by ‘Lab 1’. (**b**) Shifting of 62 dilution curves from (**a**) with respect to the antibody level to the WHO IS yields a single normalized master curve of MFI as a function of the normalized antibody concentration. Note that the axes in (**b**) were switched to allow the mathematical calculation of the optimization functions.

**Figure 2 ijms-24-15705-f002:**
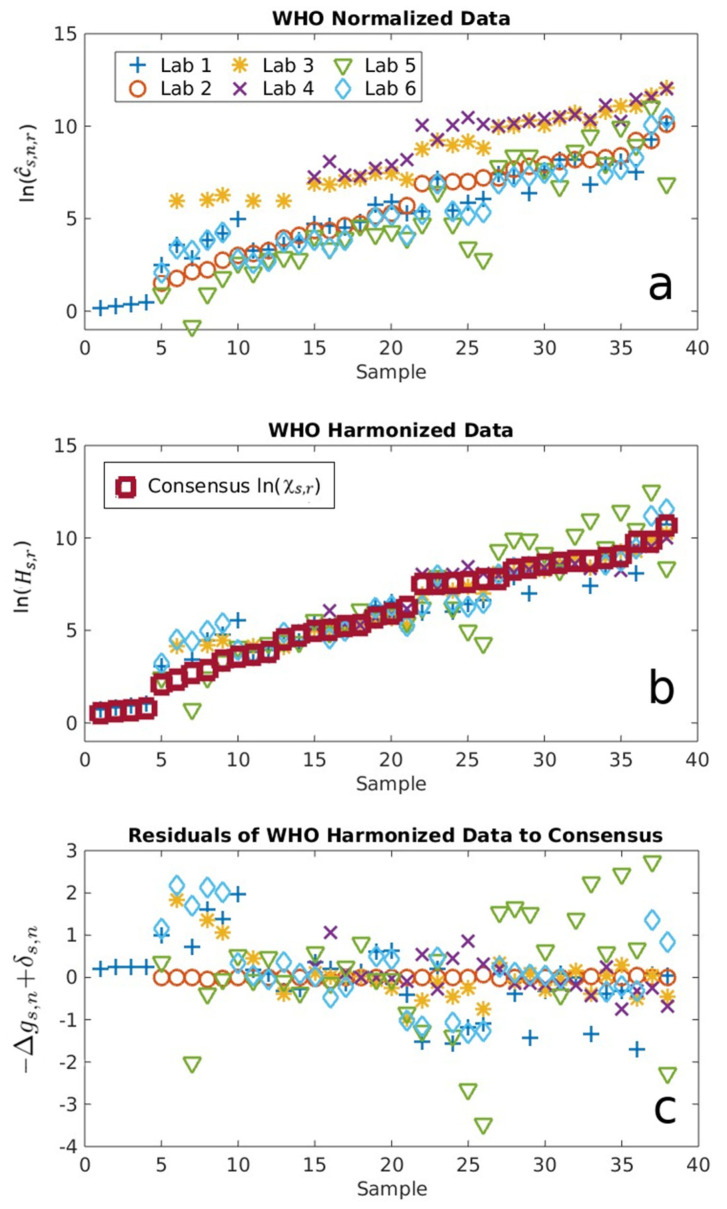
(**a**) Log-normalized antibody concentrations ln(${\hat{c}}_{s,n,r}$), normalized to the WHO IS for 38 positive convalescent serum samples determined by the six different serological IgG assays labeled as Labs 1–6; (**b**) log-harmonized antibody concentrations $H_{s,r}$ shows a greater concordance with ln(c_s,r_), shown as red squares; and (**c**) assay- and sample-dependent uncertainties, $-\Delta g_{s,n}+\delta_{s,n}$, which reflect the randomness of each individual’s sample–assay interaction.

**Figure 3 ijms-24-15705-f003:**
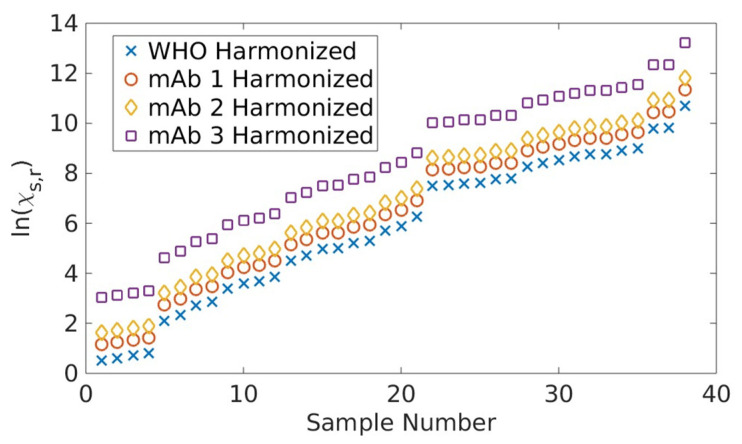
Log consensus antibody concentration values associated with each antibody standard. The estimates are ordered by increasing value according to the WHO IS-harmonized measurements. Note that, while the consensus values depend on the reference material, all consensus values for a given reference differ by the same constant relative to another reference across the samples.

**Figure 4 ijms-24-15705-f004:**
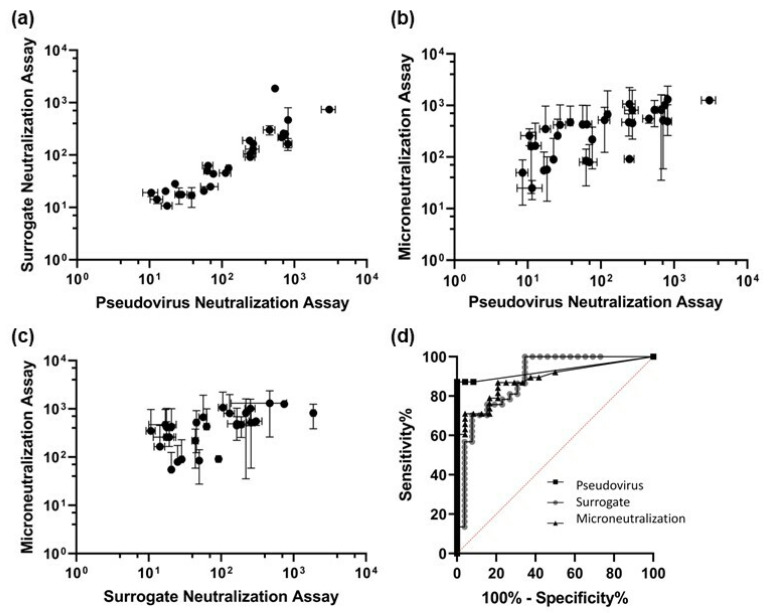
Wilcoxon matched-pairs signed-rank tests were performed comparing the pseudovirus-based neutralization assay (pvNT) to surrogate neutralization assay ((**a**) *n* = 28, *p* < 0.0001), pvNT to live virus-based microneutralization assay (MN) ((**b**) *n* = 34, *p* < 0.0001), and surrogate assay to MN ((**c**) n = 28, *p* < 0.0001). The error bars are the standard deviations obtained from the sample replicates (*n* ≥ 3). (**d**) Comparison of the receiver operating characteristic (ROC) curves and the area under curve (AUC) for three different neutralization assays used in the study.

**Figure 5 ijms-24-15705-f005:**
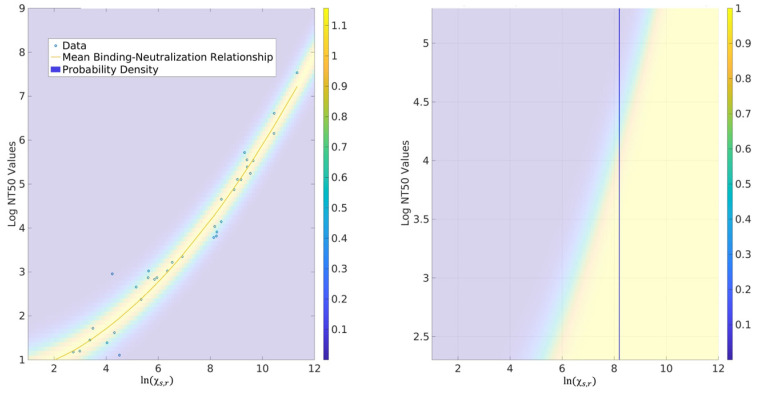
**Left**: Consensus-binding concentrations ln(c_s,r_) associated with mAb 1 compared to the NT50 values associated with pvNT from Lab 1. Data presented are from 37 positive samples. **Right**: Integral of the probability function shown in the left figure is plotted. The vertical line in the right plot is the binding level (~8.1), for which there is a >95% probability that the neutralizing level is greater than 3.68 (Ln40). Color bars adjacent to the plots indicate the numerical scales of the probabilities. See the Results and Discussion Sections for additional context.

**Figure 6 ijms-24-15705-f006:**
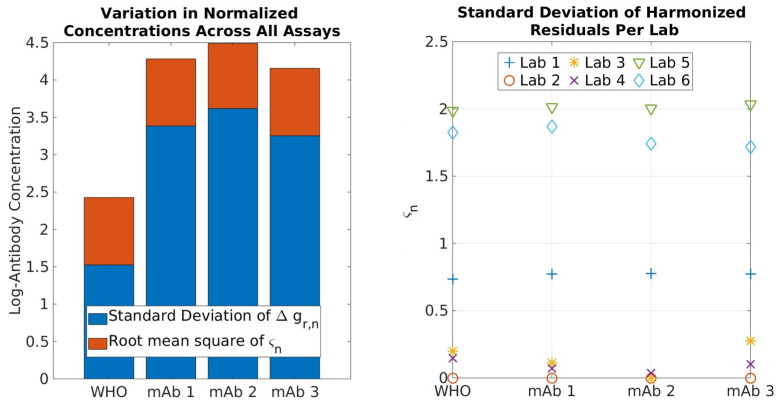
**Left**: Contribution of bias from $\Delta g_{r,n}$ and $\Delta g_{s,n}$ across all assays using each reference material. Note that, although $\Delta g_{r,n}$ and $\Delta g_{s,n}$ carry a sign in Equation (7), these show their contributions in absolute values. **Right**: Average residuals of the bias-corrected antibody concentrations for each lab and reference material.

**Table 1 ijms-24-15705-t001:** Serological and neutralization assays used in this interlaboratory study.

Participants	Serological Binding Assay(s)	Neutralization Assay(s)
NIST	SARS-CoV-2 spike IgG assay (quantitative, Wuhan-Hu-1)	(1) Surrogate bead-based neutralization assay (2) Pseudovirus-based neutralization assay by (a) fluorescence imaging and (b) flow cytometry (Wuhan-Hu-1, HEK-293 hAce2-TMPRSS2-mCherry)
FDA	(1) SARS-CoV-2 spike IgG assay (qualitative, Wuhan-Hu-1) (2) SARS-CoV-2 RBD IgG assay (qualitative, Wuhan-Hu-1)	Live virus-based microneutralization assay (D614G, Alpha, Beta, Delta)
FNLCR/NCI	(1) SARS-CoV-2 Spike IgG Assay (quantitative) (2) SARS-CoV-2 Nucleocapsid IgG Assay (quantitative)	
Abbott	ARCHITECT i2000SR Immunoassay (quantitative, Wuhan-Hu-1, spike/RBD IgG, chemiluminescent microparticle immunoassay, FDA- and EUA-granted)	
Roche	Elecsys anti-SARS-CoV-2 S assay on cobas e 801 immunoanalyzer (quantitative (US: semi-quantitative), Wuhan-Hu-1, S1 RBD total Ig, double-antigen sandwich, ligand-binding assay, FDA- and EUA-granted)	

## Data Availability

All study raw data reported by participants are saved in a shared folder of the study kept by P.N.P. and A.J.K. at NIST and are available upon request.

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
