# Peer review of "Monoclonal Antibodies as SARS-CoV-2 Serology Standards: Experimental Validation and Broader Implications for Correlates of Protection"

_ijms, 2023, doi:10.3390/ijms242115705_

Round 1
Reviewer 1 Report
Comments and Suggestions for Authors
Very interesting work, with good narrative as well as well presented.
Timely enough for its publication.
The following are additional comments:
The main question that the authors explore is developing and assessing a serology standard specific for SARS-CoV-2.
In my opinion, it is relevant for two reasons:
Serological assays are still more common than PCR to diagnose COVID-19.
While COVID-19 is not a sanitary emergency anymore, it is still a disease present across the world. Assessing the protection granted by vaccinations will be an ongoing task.
It is a very well organized effort into serology and its validation.
It would be important to move the study into real-world testing, but that could be another study.
The figures are complex and require the use of color, but overall they are well organized and can be understood.
Author Response
Please see attachment. We thank the reviewer for reading the manuscript.

Reviewer 2 Report
Comments and Suggestions for Authors
The manuscript by Wang et al establishes mathematical models (equations) to define normalization and harmonization of serological assays results and uses these (and 6 more equations) to evaluate the normalisation and harmonization of the potency of 62 SARS-CoV-2 clinical samples using the WHO International Standard and 4 monoclonal antibodies in a multi lab collaborative study, coming to the conclusion that mAbs can work as well as a pool of convalescent plasma (the WHO IS).
First of all, the terms in the equations are well explained and can be followed, but I cannot comment on whether they are valid or not and their biological significance. As a reader I have to accept them and I struggled to follow the manuscript and being convinced by the data, because of that complexity. A reviewer with a better grasp on mathematical models and above all their relevance in the biological settings, will be in a better position to express an expert opinion.
The following comments are based on what I have understood, and they are all minor points:
1) I agree with the definitions and distinction between the concept of “normalisation” and “harmonization” ; I have spotted one “standardization” word in line 447 and as this is a further add on to the use of standards (i.e. same SOPS, sharing critical reagents) I would remove it.
2) One of the limits of the use of mAb as standard is the immune evasion by the virus which makes the standard easily not relevant, as shown by therapeutic antibodies Even for convalescent plasma the WHO IS which was a pool was much more tolerant to variants than plasm from single individual. A cocktail of mAbs with different specificities could be a better alternative. The ability of a cocktail of mAbs to normalised results from different labs in a comparable level to a pool of plasma has been shown in the collaborative study for the development of WHO IS for Lassa virus antibody (WHO/BS/2021.2406: WHO 1st International Standard - Lassa fever virus antibody).
Long comment just to say that in the conclusion the limit of using a mAb as reference reagent should be addressed.
3) I don’t agree in dividing the type of assays as “serology assays” and “neutralisation assays”; as serum and plasma as source of antibodies are used in the neutralisation assays those are indeed serology assays. As per table 1 I would call them Binding assays.
4) Line 83-84 that is not true. The first WHO International Standard for antibodies to SARS-CoV-2 variants of concern has been recommended for use for new variants and has activities against recent variants such as BQ1 and XBB.
5) Line 71 an “international” between WHO and standard is missing
6) Line 57...what are the “two” CoPs? Reading the paragraph above I can only find neutralising antibody titres mentioned. Also a reference to the statement will be helpful.
Author Response
Please see the attachment, we thank the reviewer for reading the manuscript
